# Comparison of Industrial and Lab-Scale Ion Exchange for the DeNOx-SCR Performance of Cu Chabazites: A Case Study

Valentina Rizzotto [1,2], Stefan Bajić [1,2], Dario Formenti [1], Xiaochao Wu [1,2,*], Silke Sauerbeck [3], Jonas Werner [4], Thomas E. Weirich [4], Tobias Janke [2,5], Peter Mauermann [2,6], Stefan Pischinger [2,6], Regina Palkovits [2,5] and Ulrich Simon [1,2,*]

1 Institute of Inorganic Chemistry, RWTH Aachen University, Landoltweg 1a, 52074 Aachen, Germany
2 Center for Automotive Catalytic Systems Aachen, RWTH Aachen University, 52062 Aachen, Germany
3 Clariant Produkte (Deutschland) GmbH, Waldheimer Str. 15, 83052 Bruckmuehl-Heufeld, Germany
4 Central Facility for Electron Microscopy, RWTH Aachen University, Ahornstraße 55, 52074 Aachen, Germany
5 Institute of Technical and Macromolecular Chemistry, RWTH Aachen University, Worringerweg 2, 52074 Aachen, Germany
6 Institute of Thermodynamics, RWTH Aachen University, Forckenbeckstraße 4, 52072 Aachen, Germany
* Correspondence: xiaochao.wu@ac.rwth-aachen.de (X.W.); ulrich.simon@ac.rwth-aachen.de (U.S.)

**Abstract:** The efficiency and robustness of selective catalytic reduction (SCR) by $NH_3$ catalysts for exhaust gas purification, especially of heavy-duty diesel engines, will continue to play a major role, despite the increasing electrification of powertrains. With that in mind, the effect of the synthesis scale on commercially available Cu-exchanged chabazite catalysts for SCR was investigated through physicochemical characterizations and catalytic tests. During hydrothermal aging, both industrial and lab-scale prepared catalysts underwent structural dealumination of the zeolite framework and redistribution of the Al sites. Although both catalysts demonstrated similar NO conversion activity under SCR conditions, the lab-scale catalyst showed higher selectivity and lower activity in $NH_3$ oxidation. Variations in $N_2O$ formation and $NH_3$ oxidation rate were found to correlate with the formation of different copper species, and the compositions become less controllable in industrial-scale process. This case study focused on routes of ion exchange, and the results provide new insights into catalytic performance of the industrially-produced zeolites.

**Keywords:** DeNOx; $NH_3$-SCR; Cu-zeolite; ion exchange; Cu source

## 1. Introduction

Nitrogen oxides (NOx), as primary pollutants from lean burning engines, remain a challenge for exhaust gas aftertreatment. One of the currently most efficient approaches in the field is the selective catalytic reduction (SCR) of NOx. In this kind of DeNOx aftertreatment, a reducing agent is introduced in the exhaust mixture to convert NOx to harmless $N_2$ and $H_2O$, supported by a suitable catalyst [1–5]. Zeolites are a group of crystalline inorganic materials with regular pore structures that consist of connected $TO_4$ (T represents the framework atom) tetrahedra sharing oxygen atoms [2,6]. In the most common systems of $NH_3$-SCR, the catalyst of choice is a copper-exchanged zeolite. More specifically, copper-exchanged small-pore chabazite (CHA) zeolites are currently the leading $NH_3$-SCR catalysts due to their high NO conversion rates over a wide range of temperatures, high selectivity towards the formation of $N_2$ and long stability under operative SCR conditions [7–10].

In the synthesis of CHA zeolite, the temperature, seeds and templates all influence the space-time yields and catalytic properties. Many efforts have been devoted to obtaining optimized CHA zeolites by adjusting the synthetic parameters. General synthetic strategies include using different templates [11,12], controlling the distribution of Al or Si atoms [13–15], one pot synthesis methods [16,17] and microwave-assisted synthesis [18,19].

However, plenty of work has focused on improving the intrinsic properties of the CHA zeolite framework, without considering the suitability for their industrial applications, which are typically guided by economic and ecologic constraints. The large-scale synthesis conditions and the subsequent Cu ion exchange step may influence the properties of CHA zeolites, especially the hydrothermal stability [20].

In this work, we showcase the comparison of an industry-scale and a lab-scale version of Cu ion exchange process in terms of the catalytic properties of Cu-exchanged chabazites. By means of thorough physicochemical characterization, the differentiating parameters were identified, allowing us to interpret the differences in the catalytic behavior. Taking economic aspects into account, we compared two routes of ion exchange for the catalyst fabrication, i.e., a large scale or industrial process, and a small scale or laboratory process, which differ mainly in Cu source and reaction temperature. Both catalysts were prepared by performing aqueous ion exchange on a commercially available $NH_4$-form chabazite. Each aged material underwent a full physicochemical characterization routine, including inductively coupled plasma–optical emission spectroscopy (ICP-OES), powder X-ray diffraction (pXRD), diffuse reflectance infrared Fourier-transform spectroscopy (DRIFTS), $N_2$ adsorption, scanning and transmission electron microscopy (SEM and TEM) and solid-state $^{27}Al$ and $^{29}Si$ magic angle spinning nuclear magnetic resonance (NMR) spectroscopy. The as-synthesized Cu-exchanged chabazites are denoted as Cu-CHA-I (I = "industrial") and Cu-CHA-L (L = "laboratory"), respectively. To test their stability and properties under conditions similar to the operative ones, both materials were hydrothermally annealed, and the corresponding samples are deonted as Cu-CHA-I-a and Cu-CHA-L-a, respectively.

## 2. Results

The Cu/Al molar ratio was determined by ICP-OES, and the Cu loading of the two catalysts matched each other quite well (cf. Table 1). Even after hydrothermal aging, the Cu/Al ratio remained substantially unchanged in both materials. Comparatively, Cu-CHA-I and Cu-CHA-I-a showed higher Cu weight percentage values. The pXRD patterns (Figure S1) confirmed the stability of the chabazite crystalline structure not only directly after the ion exchange, but also after the hydrothermal treatment. From the combination of pXRD and DRIFTS analysis (Figure S2), no evidence was found for the presence of crystalline copper oxides particles or agglomerates ($CuO_x$) [21]. Compared with the starting material, i.e., the bare chabazite, the intensities of asymmetric vibrations of the zeolite framework (T-O-T, 1350–950 cm$^{-1}$) were decreased in the aged materials, which is in line with previous findings and points at partial lattice degradation [22]. This effect was slightly more pronounced in the industrial chabazite. Cu-CHA-I and Cu-CHA-L showed the same microscopic morphology when observed by SEM (Figure S3), and both materials appeared to be reduced to smaller particles after aging. A closer observation with TEM (Figure S4) revealed that the lab-scale catalyst better retained the particle morphology after aging: Cu-CHA-L-a presented cubical particles with pronounced facets, whereas these were less defined in Cu-CHA-I-a, and the latter's surface was partially decomposed into smaller amorphous particles. TEM scanning electron nanodiffraction (SEND) and subsequent analysis by automatic crystal orientation mapping (ACOM) [23] realized with the ASTAR system was employed to determine particle crystallinity from so called "index maps". Index maps depict the template matching quality in greyscale. SEND patterns from crystalline regions generate a higher index score, and therefore, will appear bright, whereas diffuse scattering from non-crystalline regions generates low index values, and therefore, will appear darker. Difference images (Figure 1) were constructed from index maps superimposed on virtual bright field maps (VBF, greyscale from mass/thickness + diffraction contrast) revealed that the aging process induced the formation of amorphous regions in the zeolite particles of Cu-CHA-I-a (indicated by d1, d2 and red arrows in Figure 1d). In contrast, the Cu-CHA-L-a sample did not show completely amorphized regions, although low crystallinity regions (indicated by b1 and b2 in Figure 1b) were occasionally observable. Moreover, the Z-contrast images (Figure 2) evidenced in both

zeolites the presence of smaller (4 nm–6 nm) evenly distributed particles, which were identified as non-crystalline $CuO_x$. $N_2$ adsorption measurements were performed to investigate the porosity of the two samples. The shape of the adsorption isotherms curves, as shown in Figure S5, confirms the typical microporous structure of zeolites, and Cu-CHA-I-a and Cu-CHA-L-a presented the specific BET surface areas of 633 and 606 $m^2\,g^{-1}$, and the total pore volumes of 0.303 and 0.290 $cm^3\,g^{-1}$, respectively. In general, the two samples showed similar porosity, as expected for catalysts derived from the same source zeolite ($NH_4$-form CHA), indicating that their microporous structure was only mildly affected by the Cu-exchange and the hydrothermal aging steps. The slightly higher BET surface area and accessible pore volume of Cu-CHA-I-a might be attributable to the less defined crystal facets which were observed by TEM in Figure S4. Overall, the two catalysts showed very similar physicochemical properties, even though the lab-scale material appeared to be less affected by the aging treatment.

**Table 1.** Chemical composition determined by ICP-OES.

| Sample | Si [wt%] | Al [wt%] | Cu [wt%] | Cu/Al Molar Ratio |
|---|---|---|---|---|
| Cu-CHA-I | 38.40 | 5.40 | 2.94 | 0.231 |
| Cu-CHA-L | 22.99 | 4.09 | 2.29 | 0.238 |
| Cu-CHA-I-a | 35.88 | 6.18 | 3.62 | 0.249 |
| Cu-CHA-L-a | 29.18 | 5.04 | 2.86 | 0.241 |

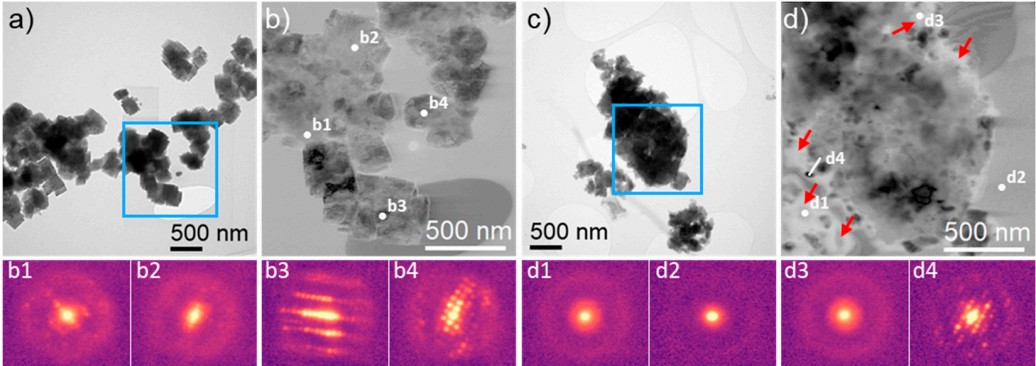

**Figure 1.** TEM bright field images (TEM-BF, (**a**,**c**)) for reference and corresponding difference images (index map superimposed on VBF, (**b**,**d**)) of Cu-CHA-L-a (**a**,**b**) and Cu-CHA-I-a (**c**,**d**). Difference images show enlarged regions marked by the reticule in (**a**,**c**). Local nanodiffraction patterns (false colored) acquired by SEND of Cu-CHA-L-a (**b1**–**b4**) and Cu-CHA-I-a (**d1**–**d4**) correspond to positions indicated by the markers in (**b**,**d**). Completely amorphized regions visible in Cu-CHA-I-a material, as shown in (**d**) (red arrows and exemplarily marked with (**d1**,**d3**)), are distinguishable by bright contrast in the difference image, whereas comparatively bright contrast regions in (**b**) (exemplarily marked with (**b1**,**b2**)) show partially amorphized/low crystallinity Cu-CHA-L-a. A nanodiffraction pattern of the TEM carbon support film is shown for reference (**d2**). Crystalline regions are distinguishable by darker contrast in the difference images, as referenced by crystalline nanodiffraction patterns (**b3**,**b4**) for Cu-CHA-L-a and (**d4**) for Cu-CHA-I-a.

The SCR performances of the two Cu-exchanged zeolites were compared in terms of $NH_3$-uptake, NO conversion, $N_2O$ formation and $NH_3$ oxidation under SCR reaction conditions.

The SCR catalytic cycle in Cu-chabazites is initiated by the reaction of NO with $NH_3$, which is pre-adsorbed at the $Cu^{2+}$ active sites. Therefore, the amount of $NH_3$ that a zeolite is capable of adsorbing and the fraction that binds to $Cu^{2+}$ sites are key parameters for the choice of an SCR catalyst [21]. The $NH_3$ storage capacities of Cu-CHA-I-a and Cu-CHA-L-a were derived from temperature-programmed desorption ($NH_3$-TPD) experiments, and the respective data are summarized in Table 2. As shown in Figure 3, the curve deconvolution

exhibits that the mid-temperature (MT, ca. 250 °C) peak, which is attributed to the $NH_3$ bound to extra-framework metal sites [24,25], is two times more intense for Cu-CHA-I-a as for Cu-CHA-L-a. This result is consistent with the 25% higher mass of Cu present in Cu-CHA-I-a, which was proved by ICP, since four-fold planar $[Cu(NH_3)_4]^{2+}$ complexes are the most stable species formed by extra-framework $Cu^{2+}$ in the presence of $NH_3$. Higher amounts of $[Cu(NH_3)_4]^{2+}$ complexes may also serve as anchor points for the formation of weakly bound $NH_3$ chains [26], which are stable at low temperatures (LT peak, ca. 190 °C). The amounts of free Brønsted sites (high temperature, HT peak, 340–350 °C) of the two aged zeolites appeared to be very similar [27]. Overall, the industrial catalyst had a 50% higher $NH_3$ storage capacity than the lab-scale one, which is mainly related to the higher Cu content.

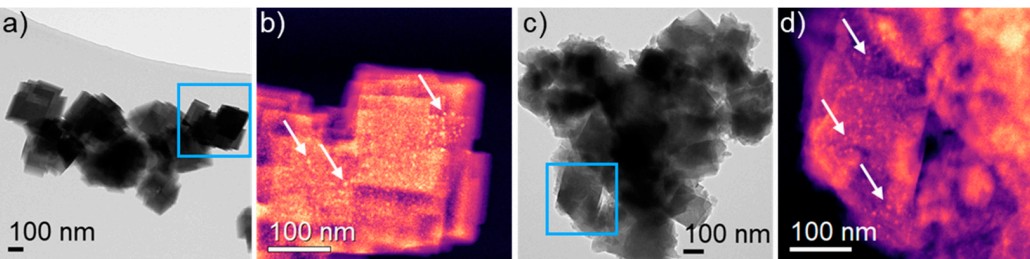

**Figure 2.** TEM bright field images (TEM-BF, (**a**,**c**)) for reference and corresponding Z-contrast images ((**b**,**d**), false colored) of Cu-CHA-L-a (**a**,**b**) and Cu-CHA-I-a (**c**,**d**). Z-contrast images show enlarged regions marked by the reticule in (**a**,**c**). Example specks of $CuO_x$ (particle size 4–6 nm) residing on CHA-material are marked with white pointers.

**Table 2.** $NH_3$ uptake derived by deconvolution of the $NH_3$-TPD profile in the low, mid and high temperature ranges (LT, MT and HT, respectively).

| NH₃ Uptake | Cu-CHA-I-a | Cu-CHA-L-a |
|---|---|---|
| Total [mmol/g] | 0.682 | 0.430 |
| LT range [mmol/g] | 0.155 | 0.096 |
| MT range [mmol/g] | 0.371 | 0.181 |
| HT range [mmol/g] | 0.163 | 0.147 |

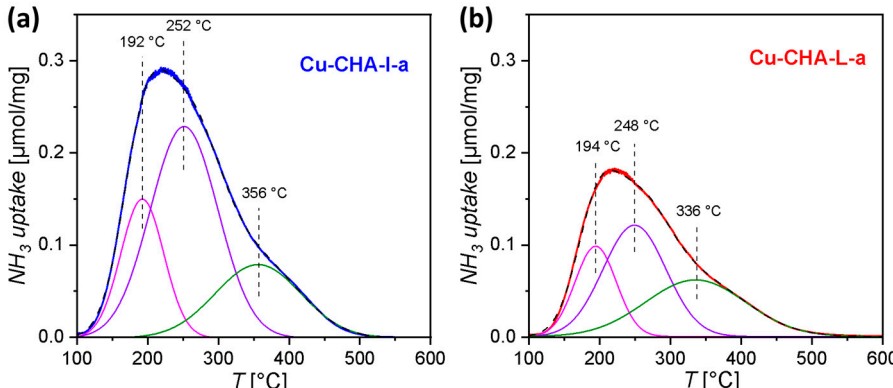

**Figure 3.** Deconvoluted $NH_3$-TPD profiles of (**a**) Cu-CHA-I-a and (**b**) Cu CHA-L-a.

The higher $NH_3$ storage of Cu-CHA-I-a seems to be beneficial for the catalytic conversion of NO at low temperatures (Figure 4): under typical SCR reaction conditions, the industrial Cu-chabazite reaches complete NO conversion at a lower temperature (200 °C) and shows ca. 15% higher conversion than Cu-CHA-L-a between 150 and 175 °C. Above 250 °C, this advantage is no more remarkable, and the two catalysts behave similarly.

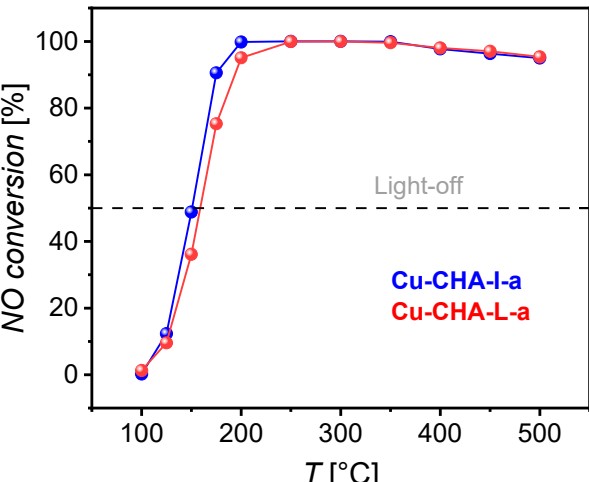

**Figure 4.** NO conversion, under $NH_3$-SCR conditions, of Cu-CHA-I-a (blue) and Cu CHA-L-a (red).

However, NO conversion cannot be utilized as the only parameter, as selectivity has to be taken in account as well.

The most harmful by-product of SCR is $N_2O$, which is a potent greenhouse gas [7]. Recent studies showed that, for Cu-exchanged zeolites, $N_2O$ may be formed not only when $NO_2$ is the predominant $NO_x$ species, but also under standard SCR conditions due to non-selective SCR [28,29]:

$$4NO + 4NH_3 + O_2 \rightarrow 4N_2O + 3H_2O. \tag{1}$$

Considering the selectivity towards $N_2O$, the two investigated catalysts showed significantly different performance. Cu-CHA-I-a produced more $N_2O$ in all measured temperature ranges (Figure 5a).

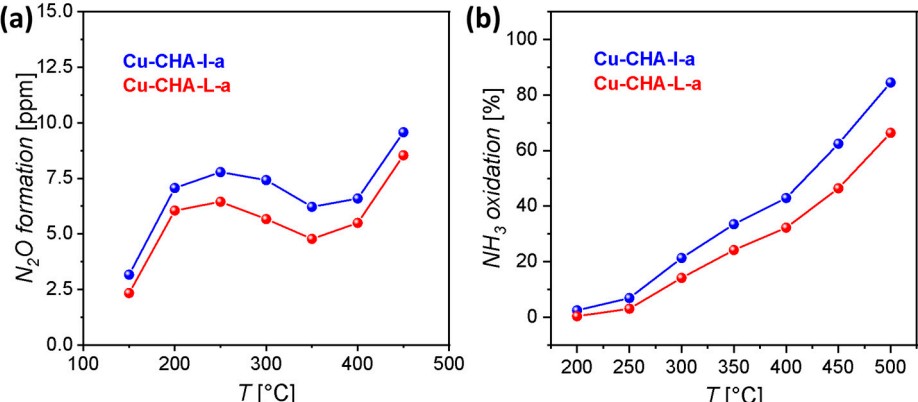

**Figure 5.** Investigation of SCR side reactions: $N_2O$ formation under (**a**) $NH_3$-SCR conditions and (**b**) $NH_3$ oxidation in the absence of NO for Cu CHA I-a (blue) and Cu CHA-L-a (red).

Another important side reaction that affects the SCR efficiency at high temperatures (>300 °C) is the oxidation of $NH_3$ (Equation (2)):

$$4NH_3 + 5O_2 \rightarrow 4NO + 6H_2O \tag{2}$$

By excluding NO from the reaction mixture, we performed specific $NH_3$ oxidation experiments on Cu-CHA-I-a and Cu-CHA-L-a. The results indicated that the industrial catalyst is more active in $NH_3$ oxidation than the lab-scale material (Figure 5b), and this is even more pronounced at higher temperatures. Cu-CHA-L-a was therefore proven to

be the catalyst with the higher selectivity towards SCR conversion of NO, even though its activity is slightly less than that of Cu-CHA-I-a at low temperatures.

The loss of crystallinity in the industrial chabazite caused by the aging process indicated faster decay of its catalytic performance. However, the change in selectivity must be related to differences in the speciation of the Cu redox-active sites. Gao et al. [29] proved that a crucial factor in the changes in selectivity, and specifically in the suppression of $NH_3$ oxidation, is the predominance of isolated Cu sites. The $Cu^{2+}$ ions coordinated to the zeolite framework in proximity of a chabazite 8-membered ring hosting a single Al atom are indicated as "ZCuOH". Due to the coordination of a hydroxyl group, these Cu active sites are known to be easier to reduce than the isolated Cu sites ($Z_2Cu$), which are located on the 6-membered ring occupied by two Al atoms [30]. Therefore, we performed temperature programmed reduction experiments with $H_2$ ($H_2$-TPR) to investigate the Cu speciation in the two catalysts. As shown by the intense peak at 260–280 °C, this analysis (Figure 6) confirmed that the predominant Cu species in Cu-CHA-I-a is ZCuOH. On the other hand, Cu-CHA-L-a shows a more even distribution of the Cu species between ZCuOH and $Z_2Cu$ (350 °C–400 °C). Moreover, a peak at ca. 110 °C is in the Cu-CHA-I-a profile, which may be related to the formation of amorphous oxide species during the aging process. Considering these results and the outcome of the physicochemical characterization, the treatment experienced by the starting material during the industrial ion-exchange may cause a redistribution of the Al sites, and therefore, the change in the Cu speciation, probably due to processes of dealumination and re-insertion.

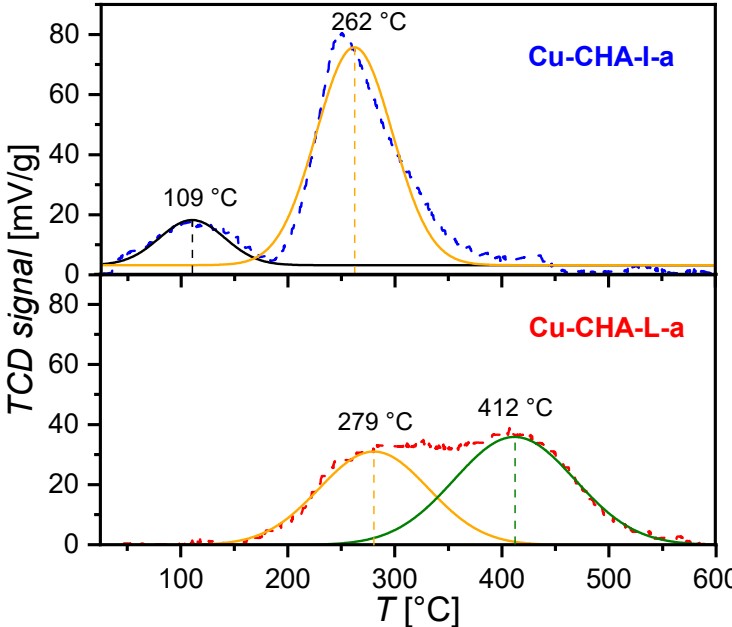

**Figure 6.** Deconvoluted $H_2$-TPR profiles of Cu-CHA-I-a (**top**) and Cu CHA-L-a (**bottom**).

To investigate the change in Al local environment in the zeolite framework induced by hydrothermal aging, solid-state $^{27}$Al and $^{29}$Si magic angle spinning NMR spectroscopy were conducted for the as-synthesized and aged Cu-chabazites. As shown in Figure 7a, both Cu-CHA-I and Cu-CHA-L showed a primary peak—"*"—at chemical shift 58 ppm and a low intensity peak—"ˆ"—at 0 ppm, which were attributed to tetrahedral Al incorporated into the framework and extra-framework octahedral Al, respectively [31]. The intensity of Cu-CHA-L is much higher than that of Cu-CHA-I, indicating the loss of crystallinity of the industrial chabazite. After being hydrothermally aged, the peak of tetrahedral Al tended toward a lower chemical shift, and the intensity decreased, whereas the peak of octahedral Al became broader and tended toward a higher chemical shift. This phenomenon is typical for hydrothermally aged Cu zeolite, which indicates the loss of Brønsted acid sites and

dealumination of the zeolite framework [4]. The elution of aluminum from the framework and the formation of $Al_2O_3$ clusters small in size are evident from the high fraction of pent-coordinated Al sites, as was proven recently [32]. The intensity ratio of the above two peaks was calculated to estimate the contents of different Al sites, which are shown in Table S1. The peak intensity ratio of Cu-CHA-L was 5.76, which was slightly higher than that of Cu-CHA-I (5.39), indicating a higher number of framework Al sites, and thus higher crystallinity for lab-scale zeolite [33]. The corresponding $^{29}Si$ NMR spectra are displayed in Figure 7b. Cu-CHA-I and Cu-CHA-L contained two framework tetrahedral Si features. The peaks at $-105$ and $-111$ ppm were attributed to tetrahedral Si with three Si neighbors and one Al neighbor (i.e., $Si(OSi)_3(OAl)$) and the same with four Si neighbors (i.e., $Si(OSi)_4$), respectively [34,35]. After hydrothermal aging, the peak at $-105$ ppm decayed, which is consistent with dealumination in zeolite framework shown in Figure 7a.

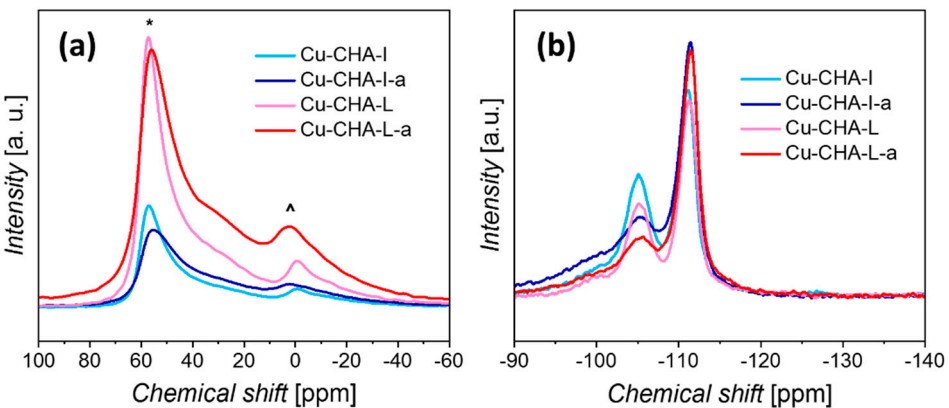

**Figure 7.** (**a**) $^{27}Al$ and (**b**) $^{29}Si$ solid state NMR spectra of Cu-CHA-I, Cu-CHA-I-a, Cu-CHA-L, Cu-CHA-L-a. In $^{27}Al$ NMR spectra, the peaks marked with "*" and "^" were attributed to tetrahedral Al incorporated into the framework and extra-framework octahedral Al, respectively.

## 3. Materials and Methods

### 3.1. Experimental Section

3.1.1. Synthesis via Liquid Ion-Exchange (LIE)

Cu-CHA-L: As a precursor material for the catalyst, a commercially available chabazite in the $NH_4$-form was supplied by Clariant (sample code: $NH_4$-CZC-13). The Si/Al ratio of the zeolite was 12.88 (measured by ICP-OES). The precursor material was modified by means of LIE, according to a procedure reported in the literature, and subsequently optimized [36]. In particular, the reaction temperature was increased, in order to reach higher Cu-exchange levels [37]. The precursor zeolite (5 g/600 mL of solution) was finely ground with a mortar and pestle and stirred under reflux in an aqueous solution of $Cu(CH_3COO)_2 \cdot H_2O$ (Honeywell Riedel-de-Haën, Selze, Germany, 99.9%). A series of Cu-exchanged chabazites were prepared by varying the reaction temperature (from room temperature to 80 °C), stirring time (up to 24 h) and $Cu^{2+}$-source concentration (from $1.61 \times 10^{-3}$ M to $9.66 \times 10^{-3}$ M). On certain samples, LIE was repeated up to three times. The product of each LIE process was recovered by vacuum filtration; washed with ultrapure water until the filtrate got white, then again with 300 mL of water; and finally allowed to dry with in air. To remove the possible remaining $NH_4^+$ extra-framework ions, a calcination step was performed in a Muffle furnace: the sample was heated for 1 h at 100 °C (1 h of ramp time) and for 6 h at 640 °C (1 h of ramp time).

Cu-CHA-I: Similar liquid ion-exchange was performed for the $NH_4$-CZC-13 precursor material. A slurry of 0.25 kg $Cu(CH_3COO)_2 \cdot H_2O$ (technical grade, VRW), 1.11 kg $NH_4$-CZC-13 (LOI of 10 wt.%) and 9 kg demineralized water was stirred at 40 °C for 2 h. The materials were filtered, washed and dried at 100 °C for 16 h. The exchange was repeated 2 times.

### 3.1.2. Hydrothermal Aging

Both the chabazite Cu-exchanged by Clariant (Cu-CZC-13) and the lab-scale synthesized materials were aged in crucibles under hydrothermal conditions at 800 °C for 16 h in flowing humid air (10% $H_2O$).

### 3.2. *Physicochemical Characterization*

Each material underwent a full characterization routine before and after hydrothermal aging. The Cu/Al ratio was obtained by inductively coupled plasma–optical emission spectroscopy (ICP-OES): the sample was dissolved in HF and analyzed with a SPECTROBLUE device by SPECTRO Analytical Instrument, GmbH (Kleve, Germany). The crystalline structure of each sample was determined by powder X-ray diffraction (pXRD, STOE and Cie GmbH, Darmstadt, Germany) using a STOE Stadi MP powder diffractometer by Stoe and Cie, equipped with a Cu-anode (40 kV, 30 mA) and a Ge monochromator for the generation of the Cu $K\alpha_1$ radiation (1.54059 Å). The presence of copper oxides agglomerates was investigated by means of XRD and diffuse reflectance infrared Fourier-transform spectroscopy (DRIFTS). A Vertex 70 infrared spectrometer by Bruker and a Praying Mantis mirror system by Harrick were employed to analyze the powder sample. For the multipoint Brunauer−Emmett−Teller (BET) surface area measurements, the samples were degassed under vacuum ($<10^{-3}$ mbar) at 300 °C for 10 h. Surface areas were determined by nitrogen adsorption at −196 °C using an automated gas adsorption analyzer (Autosorb iQ model 7—Quantachrome Instrument, Anton Paar, Graz, Austria). BET Surface area was calculated considering $p/p_0$ points giving the best linear fit. Data processing was performed using ASiQWin software (Version: 5.2x, Graz, Austria). $H_2$ temperature-programmed reduction ($H_2$-TPR) was performed on a ChemBET Pulsar TPR/TPD (Quantachrome Instruments, Boynton Beach, FL, USA) equipped with a TDC detector. Samples were pretreated in a 16.3 mL min$^{-1}$ He stream for 2 h at 500 °C. $H_2$-TPR was performed with 5% $H_2$ in Ar with a total flow rate of 100 mL min$^{-1}$ by heating the sample from 25 °C up to 600 °C with a heating rate of 5 K min$^{-1}$.

The morphology of the samples was studied by means of scanning and (scanning) transmission electron microscopy (SEM and (S)TEM, respectively) using a ZEISS Leo Supra 35 VP and a 200 kV FEI Tecnai F20 (FEI Europe B. V., Eindhoven, Netherlands), respectively. (S)TEM scanning electron nanobeam diffraction (SEND) and subsequent automated crystal orientation mapping (ACOM) were performed with the ASTAR system from NanoMEGAS SPRL (Brussels, Belgium) for ascertaining the local zeolite crystallinity. SEND patterns were acquired at a lateral step size of 4.5. TEM bright field (BF) images were acquired with a Veleta S04 F camera (EMSIS GmbH, Münster, Germany). STEM annular dark field (ADF) images were obtained by an ADF detector from Fischione Instruments (E.A. Fischione Instruments, Inc., Export, PA, USA). Digital Micrograph (version 2.11.1404.0, Gatan Inc., Pleasanton, CA, USA) was used for image processing [38] and analysis of TEM BF and STEM-ADF images. The $CuO_x$ particle sizes were identified by their bright contrast in the atomic number sensitive STEM-ADF (Z-contrast) images.

All solid state nuclear magnetic resonance (NMR) spectra were recorded on a Bruker AV-III-400 MHz (BioSpin GmbH, Ettlingen, Germany) using a MAS-4 mm probe and a sample rotation frequency (rf) of 12.5 kHz. The reference frequencies (i.e., 0 ppm for the chemical shift) for the nuclei were: nue_ref ($^{29}$Si) = 79.500614500 MHz and nue_ref ($^{27}$Al) = 104.26923126 MHz. The rf field strength was determined by $^{27}$Al nutation curves on an aluminum nitrate solution in water: rf = 4.5 kHz for a power of 1.5 W and rf = 45 kHz for 150 W. All other pulses were calculated based on this value. $^{27}$Al spectra were acquired after one hard-pulse (P30 = 1.89 μs; power = 150 W), showing rotational sidebands or a rotation synchronized Hahn-echo (P1 = 18.9 μs, D6 = 51.64 μs, P2 = 37.8 μs; power = 1.5 W) where the rotational sidebands are suppressed. $^{29}$Si spectra were acquired after one hard-pulse (P90 = 11.5 μs; power = 50 W; rf = 21 kHz).

### 3.3. Temperature-Programmed Desorption with NH₃ (NH₃-TPD)

The NH₃ uptake capacity of the samples was measured with TPD experiments. The zeolite powder was pressed, crushed and sieved, in order to obtain the 200–355 µm granules employed in the analysis. The total gas flow during the test was set to 300 sccm, and $N_2$ was employed as carrier gas. The amount of sample to be used was calculated to have a gas hourly space velocity of 50,000 $h^{-1}$. The granules were located in a 6 mm diameter quartz tube and blocked with quartz wool. After a 30 min pretreatment in 6.5% $O_2$ at 500 °C, the sample was cool down to 100 °C, and afterwards, exposed to 1000 ppm $NH_3$ for 3 h. The sample was flushed for 4 h in $N_2$ and then heated up to 700 °C at 5 °C/min. The downstream gas composition was measured with a Multigas 2030 gas analyzer by MKS. The temperature was controlled with a Carbolite TZF horizontal furnace and the gas composition with mass flow controllers by MKS.

### 3.4. SCR Catalytic Tests

The selective catalytic reduction (SCR) tests were carried on all investigated materials in granule form (200 µm–355 µm), using the same reactor and measuring setup as for the NH₃-TPD experiments. Prior to the measurement, the sample was pretreated for 30 min in 6.5% $O_2$ at 500 °C. After cooling the sample to 100 °C, the standard SCR mixture was applied: 500 ppm $NH_3$/500 ppm NO/6.5% $O_2$/9% $CO_2$/5% $H_2O$ with $N_2$ as carrier gas (total gas flow = 300 sccm). The water was dosed by flowing the gas stream through a bubbler. The amount of sample to be used was calculated to have an hourly gas space velocity of 50,000 $h^{-1}$. A series of increasing temperature steps between 100 and 500 °C were applied. Each step was extended till the steady state for the temperature was reached. During these experiments, the $N_2O$ formation under SCR conditions was monitored and the NO conversion was calculated according to the following formula:

$$\text{NO conv. \%} = \frac{c_{in} - c_{out}}{c_{in}} \cdot 100\% \tag{3}$$

where $c_{in}$ is the dosed NO concentration (500 ppm) and $c_{out}$ is the NO concentration measured by the gas analyzer. The amount of sample to be used was calculated to have a gas hourly space velocity of 50,000 $h^{-1}$.

### 3.5. NH₃ Oxidation Tests

The selectivity towards NH₃ oxidation was measured by performing an experimental routine similar to the one of the SCR tests, but after removing NO from the gas mixture. To reach 300 sccm of total gas flow, additional $N_2$ was dosed. All investigated materials were in granule form (200 µm–355 µm); the same reactor and measuring setup as for the NH₃-TPD and SCR experiments was used. Equation (1) was employed for the calculation of the consumed NH₃, considering $c_{in}$ as the dosed NH₃ concentration (500 ppm) and $c_{out}$ as the NH₃ concentration measured by the gas analyzer.

## 4. Conclusions

In summary, two Cu-chabazite catalysts were prepared by starting from the same $NH_4^+$-form parent zeolite and performing the Cu ion exchange in a large-scale (industrial) or in a small-scale (laboratory) procedure. Similar Cu/Al molar ratios of 0.231 and 0.238, respectively, were achieved for the materials. After hydrothermal aging treatment, TEM images showed the lab-scale catalyst retained the original particle morphology, and the orientation imaging technique revealed the formation of amorphous regions in the industrial-scale sample. The change in surface morphology generated during hydrothermal aging treatment slightly increased the accessible pores of the industrial-scale catalyst, leading to a mildly higher measured BET surface area. For both aged materials, the SCR performances were compared in terms of NH₃-uptake, NO conversion, $N_2O$ formation and NH₃ oxidation. The industrial-scale catalyst showed ca. 15% higher NO conversion at a low temperature, whereas similar NO conversions were achieved by both catalysts at

higher temperatures. However, the lab-scale material was proved as the preferable catalyst due to its lower $N_2O$ formation and reduced $NH_3$ oxidation towards SCR conversion of NO. Furthermore, $H_2$-TPR profiles indicated the lab-scale zeolite possessed less ZCuOH species, and thus more isolated Cu sites, which enhance the SCR selectivity. NMR results illustrated the dealumination of the zeolite framework and redistribution of Al during the hydrothermal aging process.

The industrial-scale catalyst appears to destabilize the zeolite crystalline structure and made it less resistant to hydrothermal aging. Such change in the crystal structure appears to affect the Al distribution on the zeolite, and consequently, the nature and reactivity of the Cu redox sites. This study illustrates that the catalytic findings achieved in the laboratory cannot easily be transferred during a scale-up, and that advanced characterization methods should be also applied to industrial research to get a better understanding of the material destined for mass production.

**Supplementary Materials:** The following supporting information can be downloaded at: https://www.mdpi.com/article/10.3390/catal12080880/s1. Figure S1: X-ray diffractograms of the fresh and aged industrial (a) and lab-scale (b) Cu-chabazites. Figure S2: DRIFT spectra of the fresh and aged industrial (a) and lab-scale (b) Cu-chabazites. Figure S3: SEM images of the (a) Cu-CHA-I, (b) Cu-CHA-I-a, (c) Cu-CHA-L and (d) Cu-CHA-L-a. Figure S4: TEM images of the (a) Cu-CHA-I-a and (b) Cu-CHA-L-a. Figure S5: Nitrogen adsorption isotherms of Cu-CHA-I-a and Cu-CHA-L-a. Table S1: The intensity ratio of peak "*" and peak "^" in $^{27}$Al solid state NMR spectra.

**Author Contributions:** Methodology, V.R. and S.B.; investigation, V.R., D.F., J.W., P.M. and T.J.; resources, S.S.; writing—original draft preparation, V.R.; writing—review and editing, X.W., U.S. and T.E.W.; supervision, T.E.W., S.P., R.P. and U.S.; funding acquisition, U.S. All authors have read and agreed to the published version of the manuscript.

**Funding:** This work was supported by the German Federal Ministry of Education and Research (BMBF) in the context of the DeNO$_x$ project (13XP5042A), and by the Deutsche Forschungsgemeinschaft (DFG, German Research Foundation) under Germany's Excellence Strategy—Cluster of Excellence 2186 "The Fuel Science Center"; ID: 390919832.

**Data Availability Statement:** Not applicable.

**Acknowledgments:** We thank Clariant for supplying the precursor materials, Gerhard Fink for performing the solid-state NMR measurements, and Horst Schulte for contributing corrections to the manuscript.

**Conflicts of Interest:** The authors declare no conflict of interest.

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
