# Peer review of "Comparison of Industrial and Lab-Scale Ion Exchange for the DeNOx-SCR Performance of Cu Chabazites: A Case Study"

_catalysts, doi:10.3390/catal12080880_

Round 1

Reviewer 1 Report

For complete characterization of the catalysts, specific surface area studies such as low-temperature nitrogen sorption should be performed. Without this chakarterist it is difficult to assess whether the results are valid. 

The publication I reviewed is missing a fundamental test, which is low-temperature  sorption of nitrogen, which determines the textural properties of the catalysts.
The absence of this fundamental characterization makes it impossible for me to evaluate the other results. This study should have been performed in order to make a fair review.
I am surprised that the authors chose to publish their results without this basic characterization.

Author Response

We thank the reviewer for the suggestion. Accordingly, we performed the N2 adsorption experiment to investigate the porosity and determine the BET surface area of two catalysts. Cu-CHA-I-a and Cu-CHA-L-a present the specific BET surface areas of 633 m² g-1 and 606 m² g-1, and the total pore volumes of 0.303 cm3 g-1 and 0.290 cm3 g-1, respectively. In general, the two samples show similar porosity, as expected for catalysts derived from the same source zeolite (NH4-form CHA), indicating that their microporous structure was only mildly affected by the Cu-exchange and the hydrothermal aging steps. The slightly higher BET surface area and accessible pore volume of Cu-CHA-I-a might be attributed to the less defined crystal facets which were observed by TEM in Figure S4. To complete the characterization and improve the quality of our manuscript, we have added the BET result and corresponding discussion section in the revision.

Reviewer 2 Report

This is a very nice and timely study from the Palkovits/Simon groups on lab-made and industrial Cu CHA catalysts and their comparison for SCR, ammonia oxidation etc. It is important because it highlights the differences between the lab made Cu-SSZ-13 crystals vs industrially made Cu-SSZ-13. It employs excellent characterization techniques to study physicochemical properties of these materials and their differences/similarities. The main difference in the low temperature activity of the commercial vs lab made Cu-SSZ-13 crystals is ascribed to higher NH3 storage in the commercial crystals due to higher Cu content. However, commercial crystals produce more unwanted NO. In summary, this nice study should be published. I just have one minor comment that I would like the authors to consider and they may want to cite this reference DOI: 10.3390/molecules27072352 . This is related to the aged samples. Recently it emerged that the reason for the specific changes in the NMR for steamed/aged zeolites is the formation of small Al2O3 clusters - the reason why in 27Al ssNMR tetrahedral site broadens together with pronounced octa formation and high amounts of penta-Al in the middle is due to the formation of Al2O3 nanocrystals. Their small size is the reason why a lot of penta Al signal is observed since the majority of atoms are on the surface and this is typical of Al2O3 nanofacets. I suggest for the authors to mention that in the text (the elution of aluminum from the framework and formation of Al2O3 clusters with small size which (size) is evident from the high fraction of penta Al, as was proven recently 10.3390/molecules27072352. 

In summary, I congratulate the authors on this elegant study and looking forward to seeing this published.

Author Response

We are grateful for this very positive comment and also for bringing the very recent and useful literature to our attention. As suggested by the referee, we added the statement and cited the literature in the corresponding NMR result section.

Reviewer 3 Report

This paper describes the application of Cu Chabazite for DeNOx-SCR. The paper is well written, and all the data and results are well described. I accept this paper after minor revision.

1.         The authors should replace the old reference papers with papers published within the last five years. It will make the paper more up-to-date.

2.         The authors should highlight the importance of their research.

3.         The abstract and conclusion parts should be revised by adding actual data and highlighting the novelty of this research.

Author Response

1. We appreciate this suggestion. We replaced most of the old references by more recent ones, but kept some of them because of the specific professionality.

2& 3. We revised our abstract and conclusion sections, accordingly, included more actual data and we emphasized the importance of our work.

Round 2

Reviewer 1 Report

The results are still incomplete. I do not understand the selection for textural testing of only 2 catalysts out of the 4 presented in the manuscript. The authors have not explained the reason for not performing tests on the other two catalysts. Especially since the authors wrote in the introduction that "Each material underwent a full physicochemical characterisation routine" which is not true. Please complete the textural characterisations.
For better visualisation of the results, please include the results in a table, e.g. Table 1 (with a change in the caption of this table).
While reading the text, I found some errors, e.g. line 299: '-3' should be in the exponent; line 303: '0' should be as subscrip. Please check the whole text.

Author Response

We thank the reviewer for the comment, although we believe that adding these data will not be relevant to understand the central scientific aspects of our work. From the perspective of the automotive catalysis field, it is common not to present detailed investigations of freshly-synthesized catalysts, but rather to focus on the aged or de-greened catalysts, which better reproduce the catalytic behavior encountered under real on-the-road conditions. In this work, we aimed to compare the effect of industrial-scale and lab-scale Cu ion exchange on the NH3 SCR performances and deliver useful new information for the real-life application of such catalysts. The use of aged-samples was therefore mandatory and corresponds to the state of the common sense in this field of research. For this reason and due to the short time available for re-submission, we chose to perform N2 physisorption on just the two aged materials, which are the actual catalysts to be compared.

We agree with the reviewer that the N2 adsorption gives important information about the textural properties of materials. However, we have already presented NH3-TPD results in this work (Figure 3), which are more informative for the catalytic applications, since they describe the uptake capacity (directly dependent on the accessible pore volume) and the adsorption sites specific to NH3, which is the reducing agent and initiator of the SCR catalytic cycle. On the other hand, the materials analyzed in this work are derived from the same “mother” zeolite (which is a commercial product, produced in large scale by the Clariant company, who co-authored this work) and therefore are expected to retain a highly similar crystalline framework (see XRD), so that N2 sorption technique may not be precise enough to provide reliable information on subtle differences in the microporous structure. This is seen in the additionally performed N2 sorption measurements, where the two aged catalysts show similar adsorption isotherms and BET surface areas (Figure S5). Therefore, in our opinion, the N2 sorption experiments are indeed a stand characterization method for zeolites, but not indispensable to understand the content of this work.

In summary, we consider additional N2 adsorption experiments for the fresh zeolites not justified for the understanding and the scientific value of our work. In order to avoid misunderstanding, we corrected the ‘each materials’ to ‘each aged material’ in the sentence in introduction section. We therefore kindly ask you to reconsider your suggestion by considering the above-mentioned reasons.

Thank the reviewer again for pointing out the typos in the manuscript, we checked the whole text and corrected them in the revision.